# DISTRIBUTED TRANSFER LEARNING FOR DEEP CONVOLUTIONAL NEURAL NETWORKS BY BASIC PROBABILITY ASSIGNMENT

**Arash Shahriari**
Research School of Engineering, Australian National University
Commonwealth Scientific and Industrial Research Organisation
`arash.shahriari@anu.edu.au;csiro.au`

## ABSTRACT

Transfer learning is a popular practice in deep neural networks, but fine-tuning of a large number of parameters is a hard challenge due to the complex wiring of neurons between splitting layers and imbalance class distributions of original and transferred domains. Recent advances in evidence theory show that in an imbalance multiclass learning problem, optimizing of proper objective functions based on contingency tables prevents biases towards high-prior classes. Transfer learning usually deals with highly non-convex objectives and local minima in deep neural architectures. We propose a novel distributed transfer learning to tackle both optimization complexity and class-imbalance problem jointly. Our solution imposes separated greedy regularization to each individual convolutional filter to make single-filter neural networks such that the minority classes perform as the majority ones. Then, basic probability assignment from evidence theory boosts these distributed networks to improve the recognition performance on the target domains. Our experiments on several standard datasets confirm the consistent improvement as a result of our distributed transfer learning strategy.

## 1 INTRODUCTION

In supervised learning, many classification algorithms assume the same distribution for training and testing data. Consequently, change of distribution requires rebuilding of the statistical models which is not always practical because of the hardship of recollecting of training data or heavy learning process. One of the solutions is transfer learning that transfers the classification knowledge into a new domain Pan & Yang (2010). This aims at learning of highly-generalized models with different probability distributions across domains to learn novel domains without labeled data Wang & Schneider (2014) Zhang et al. (2013). Here, the main challenge is to reduce the shifts in data distribution between domains by algorithms that minimize the discriminant of the domains. It is worth mentioning that this could not get rid of domain-specific variations Long et al. (2016).

Transfer learning for deep neural networks has been proved highly beneficial to boost their overall performance. Deep learning practices usually require huge amount of labeled data to learn powerful models. The transfer learning enables adaptation to a different source with small training samples. On the other hand, deep neural networks practically learn intermediate features. They could provide better transfer among domains because some of them generalize well among various domains of knowledge Glorot et al. (2011). These transferable features generally underlies several probability distributions Oquab et al. (2014) which reduce the cross-domain discrepancy Yosinski et al. (2014).

The common observation among several deep architectures is that features learned in bottom layers are not that specific, but transiting towards top layers makes them tailored to a dataset or task. A recent study Yosinski et al. (2014) of the generality or specificity of deep layers for the sake of transfer learning reveals two difficulties which may affect the transfer of deep features. First, top layers get quite specialized to their original tasks and second, some optimization difficulties rise due to the splitting of the network between co-adapted layers. In spite of these negative effects, it

---

**Algorithm 1** Basic Probability Assignment (BPA)

 **Input:** train/validation set $\mathcal{X}$
 **Output:** basic probability assignment $\mathbf{BPA}(\phi)$

 1: compute $\mathbf{R}(\phi)$ and $\mathbf{P}(\phi)$ (Eqs.1- 2)
 2: calculate recall and precision assignments(Eq.3)
 3: apply Dempster rule for accumulation (Eq.4)

---

is shown that transferred features not only perform better than random ones but also provide better initialization. This gives a boost to the generalization of deep neural networks as well.

In this paper, we propose a framework for distributed transfer learning in deep convolutional networks. This tries to alleviate the burden of splitting networks in the middle of fragile co-adapted layers. The intuition is that above difficulty relates to the complexity of deep architectures and also, class-imbalance in the transferred domain.

On the matter of network complexity, we argue that the splitting of layers leads to a hard optimization problem because of high complexity in the interconnections between neurons of co-adapted layers. It seems that transfer learning is not able to thoroughly reconstruct the original powerful wiring for the transferred domain. This is due to the size of network and large number of interconnections across neurons. To address this issue, we fine-tune the convolutional filters separately and hence, reduce the complexity of the non-convex optimization.

On the other hand, it seems that the class-imbalance problem rises form different distribution of data in original and transferred domains. This issue can be handled by cost-sensitive imbalanced classifications methods. By class-imbalance in transferred domain, we mean variable coverage of common classes in this domain and the ones from the original domain. It is probable that both original and transferred datasets have uniform distributions of data among their classes, but some classes in one domain may be fully or partly covered by the other domain. This results in imbalance class distribution in the transfer learning.

The determination of a probabilistic distribution from the confusion matrix is highly effective to produce a probability assignment which contributes to class-imbalance problems. This basic probability assignment can be either constructed from recognition, substitution and rejection rates Xu et al. (1992) or both precision and recall rates of each class Deng et al. (2016). The key point is harvesting of maximum possible prior knowledge provided by the confusion matrix to overcome the imbalance classification challenge.

Since the power of deep convolutional models come from mutual optimization of all parameters, we join the above distributed fine-tuned filters by a boosting scheme based on basic probability assignment. Our experiments confirm the functionality of our distributed strategy for deep transfer learning. The rest of paper is organized as follows. We present the formulation of our method in Section 2, report our experiments in Section 3 and conclude in Section 4.

## 2 FORMULATION

In general, a confusion matrix represents the class-based predictions against actual labels in form of a square matrix. Inspired by Dempster−Shafer theory, construction of basic probability assignment (BPA) Sentz & Ferson (2002) gives a vector which is independent of the number of class samples and sums up to one for each individual label. This basic probability assignment provides the ability to reflect the difference contributions of a classifier to each individual classes or combine the outcomes of multiple week classifiers.

### 2.1 BASIC PROBABILITY ASSIGNMENT

A raw two-dimensional confusion matrix indexed by predicted classes and actual labels provides some common measures of classification performance. They are accuracy (the proportion of the total number of predictions that were correct), precision (a measure of the accuracy provided that

a specific class has been predicted), recall (a measure of the ability of a prediction model to se-
lect instances of a certain class from a dataset) and F-score (the harmonic mean of precision and
recall) Sammut & Webb (2011).

Suppose a set of train/validation samples $\mathcal{X} = \{X_1, \ldots, X_{|\mathcal{X}|}\}$ from $\mathcal{C} = \{C_1, \ldots, C_{|\mathcal{C}|}\}$ different
classes are assigned to a label set $\mathcal{L} = \{L_1, \ldots, L_{|\mathcal{L}|}\}$ by a classifier ($\phi$) such that $|\mathcal{C}| = |\mathcal{L}|$. If
each element ($n_{ij}$) of the confusion matrix $\mathbf{C}(\phi)$ is considered as the number of samples belonging
to class $C_i$ which assigned to label $L_j$, then we can define recall ($r_{ij}$) and precision ($p_{ij}$) ratios as
follows Deng et al. (2016)

$$
\begin{aligned}
r_{ij} &= \frac{n_{ij}}{\sum_{j=1}^{|\mathcal{C}|} n_{ij}} \\
p_{ij} &= \frac{n_{ij}}{\sum_{i=1}^{|\mathcal{L}|} n_{ij}}
\end{aligned}
\tag{1}
$$

It can be seen that the recall ratio is summed over the actual labels (rows) whilst the precision ratio
is accumulated by the predicted classes (columns) of the confusion matrix $\mathbf{C}(\phi)$. Now, we are able
to define recall and precision matrices as

$$
\begin{aligned}
\mathbf{R}(\phi) &= \{r_{ij}\} \\
\mathbf{P}(\phi) &= \{p_{ij}\} \\
for\ i &\in [1 \ldots |\mathcal{L}|],\ j \in [1 \ldots |\mathcal{C}|]
\end{aligned}
\tag{2}
$$

The basic probability assignments of these matrices contain recall and precision probability elements
for each individual class $C_i$ such that

$$
\begin{aligned}
mr_i &= \frac{r_{ii}}{\sum_{j=1}^{|\mathcal{C}|} r_{ji}} \\
mp_i &= \frac{p_{ii}}{\sum_{j=1}^{|\mathcal{L}|} p_{ij}}
\end{aligned}
\tag{3}
$$

These elements are synthesized to form the final probability assignments representing the recogni-
tion ability of classifier $\phi$ to each of the classes of set $\mathcal{C}$

$$
m_i = mr_i \oplus mp_i = \frac{mr_i \times mp_i}{1 - \sum_{i=1}^{|\mathcal{C}|} mr_i \times mp_i}
\tag{4}
$$

Here, operator $\oplus$ is an orthogonal sum which is applied by Dempster rule of combination Sentz
& Ferson (2002). The overall contribution of the classifier $\phi$ cab be presented as a probability
assignment vector

$$
\begin{aligned}
\mathbf{BPA}(\phi) &= \{m_i\} \\
for\ i &\in [1 \ldots |\mathcal{C}|]
\end{aligned}
\tag{5}
$$

It is worth mentioning that $\mathbf{BPA}(\phi)$ should be computed by the train/validation set because we
assume that the test set does not include actual labels. Besides, combination of different classes
under vertical or horizontal categories is a common practice in visual classification. The benefit
lies in the fact that bottom layers of deep convolutional architectures make better contribution to
detect first and second order features that are usually of specific directions (vertical vs horizontal)
rather than detailed distinguished patterns of the objects. This leads to a powerful hierarchical
feature learning in the case that $|\mathcal{C}| \ll |\mathcal{L}|$. In contrast, some classes can be divided to various
sub-categories although they all get the same initial labels and hence this holds $|\mathcal{C}| \gg |\mathcal{L}|$ to take
the advantage of top layers. In the above formulation, we do not merge or divide the original setup
of the datasets under study ($|\mathcal{C}| = |\mathcal{L}|$) although it seems that our BPA-based approach is also able
to boost the trained classifiers for each of the merge/divide scenarios.

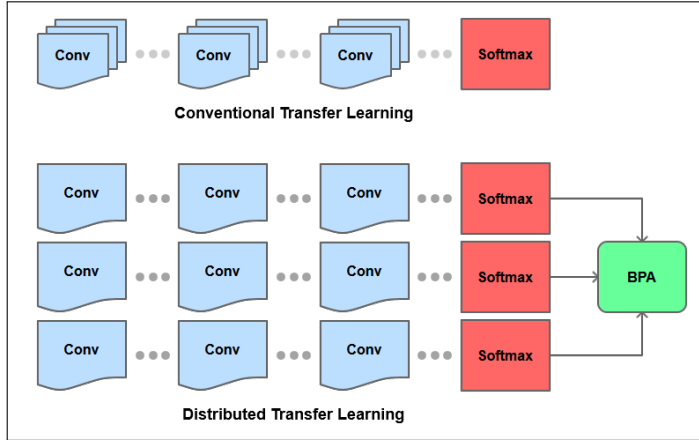

Figure 1: Conventional and Distributed Transfer Learning. The blue blocks (Conv) represent convolutional layers in the original domain, the red blocks (Softmax) show fine-tuned layers for the target domain and the green block corresponds to the basic probability assignment (BPA) respectively.

## 2.2 DISTRIBUTED TRANSFER LEARNING

A general practice in transfer learning includes training of an original deep neural network on a dataset and then, fine-tuning of learned features for another dataset on a new target network. Bengio et al. (2012). The generality of selected features for both original and target domains is critical to the success of the transfer learning. For implementation, we train the original network and copy its bottom layers to form the target network. The top layers of the target network are initialized randomly and trained on the target dataset. We are able to employ backpropagation from top to bottom layers and fine-tune their parameters for the target task or freeze the copied originals and only update top target layers. This can be decided by size of the target dataset and number of parameters in the original layers. Fine-tuning of large networks for small dataset leads to overfitting but for small network or large dataset, performance will be improved Sermanet et al. (2013).

Based on our formulation for basic probability assignment (BPA) on Section 2.1, we are able to follow the above transfer learning procedure by learning of a classifier $\phi$ (SVM or Softmax) and computing $\mathbf{BPA}(\phi)$ using Algorithm 1. Here, the learning means fine-tuning of target domain using the rained weights and biases of the original network. To implement this, we train the original fully-connected layers by the features calculated by presenting target's train set to convolutional layers of the same original network. We deploy this procedure for each of the available convolutional filters separately and compute the BPA of each individual single-filter network for train/validation sets. Then, we combine unary potentials of all the fine-tuned classifiers by employing BPA weights to come up with a unit set of class probabilities. Figure 1 provides an overview of conventional and distributed transfer learning processes.

Suppose that $C_i$ is the predicted class for a test sample $T$ provided by classifier $\phi$. To revise the classification outcome by the $\mathbf{BPA}$ calculation, we multiply the test sample's unary potentials $\mathbf{U}(T) = \{u_1, \dots, u_{|\mathcal{C}|}\}$ (probabilities of belonging to each class) by an assignment vector $\mathbf{M}(\phi) = \{1 - m_1, \dots, 1 - m_{|\mathcal{C}|}\}$ (contributions of the classifier $\phi$ to each class) and pick the maximum index as the revised predicted label

$$C(T) = \mathbb{I}\big(arg\,max\,\{u_1 \times (1 - m_1), \dots, u_{|\mathcal{C}|} \times (1 - m_{|\mathcal{C}|})\}\big) \qquad (6)$$

This implies that if classifier $\phi$ performs well on class $C_i$ (high $m_i$), it is highly probable that $C(T)$ leans towards $C_i$. At the same time, other minority classes like $C_j$ (low $m_j$) have a chance to win if their unary potentials would be high enough ($u_j > u_i$). In contrast, if $\phi$ does poor classification on class $C_i$ (low $m_i$), the possibility of updating $C(T)$ to another class ($C_j$) with even worse unary potential ($u_j < u_i$) would be higher. Therefore, $\mathbf{BPA}$ shows quite successful in handling imbalance data distribution among classes.

---

**Algorithm 2** Distributed Transfer Learning

 **Input:** train/validation set $\mathcal{X}$, test sample $T$, set of week classifiers $\mathcal{F}$
 **Output:** predicted class $C_{\mathcal{F}}(T)$

 **for** $i = 1$ **to** $|\mathcal{C}|$ **do**
 **for** $j = 1$ **to** $|\mathcal{F}|$ **do**
 1: compute $m_{ij} \in \mathbf{BPA}(\mathcal{F})$ (Alg.1)
 2: calculate unary potential $u_{ij} \in \mathbf{U}_{\mathcal{F}}(\mathbf{T})$
 **end for**
 **end for**

 3: predict boosted output $C_{\mathcal{F}}(T)$ (Eq.8)
 4: employ error backpropagation for fine-tuning

---

As described in Section 1, employing probability assignment addresses the class-imbalance problem but does not reduce the complexity of optimization because of the fact that both forward learning and error backpropagation are applied to all the model parameters. To break this non-convex optimization, we introduce our distributed transfer learning strategy. For implementation, we replace the mutual learning of all the parameters with learning of each individual convolutional filter in a separate classifier fed by the bottom original layer. It means that we train a set of week single-filter classifiers $\mathcal{F} = \{\phi_1, \ldots, \phi_{|\mathcal{F}|}\}$ which $|\mathcal{F}|$ equals the number of convolutional filters in the deep neural architecture.we follow the recipe of single classifier in Equation 5 but extend it to redefine

$$\mathbf{BPA}(\mathcal{F}) = \{m_{ij}\}$$
$$for \; i \in [1 \ldots |\mathcal{C}|], \; j \in [1 \ldots |\mathcal{F}|] \tag{7}$$

such that $m_{ij}$ is the probability assignment of class $C_i$ to week single-filter classifier $\phi_j$. To come up with class of the test sample $T$, we update the Equation 6 as follows

$$C_{\mathcal{F}}(T) = \mathbb{I}\big(arg\,max\,\{\frac{u_{1j} \times (1 - m_{1j})}{\sum_{j=1}^{\mathcal{F}} u_{1j} \times (1 - m_{1j})}, \ldots, \frac{u_{ij} \times (1 - m_{|\mathcal{C}|j})}{\sum_{j=1}^{\mathcal{F}} u_{|\mathcal{C}|j} \times (1 - m_{|\mathcal{C}|j})}\}\big) \tag{8}$$

Here, $u_{ij}$ is the unary potential of class $C_i$ determined by the week single-filter classifier $\phi_j$. Building on the above formulations, we are able to distribute the transfer learning among convolutional filters and join them later to implement a better fine-tuning for the target deep convolutional network according to the Algorithm 2.

## 3 EXPERIMENTS

We conduct our experiments on MNIST, CIFAR and Street View House Numbers (SVHN) datasets. The MNIST dataset LeCun et al. (1998) contains $60,000$ training examples and $10,000$ test samples normalized to $20 \times 20$, centered by center of mass in $28 \times 28$ and sheared by horizontally shifting such that the principal axis is vertical. The foreground pixels were set to one and the background to zero. The CIFAR dataset Krizhevsky & Hinton (2009) includes two subsets. CIFAR-10 consists of 10 classes of objects with $6,000$ images per class. The classes are airplane, automobile, bird, cat, deer, dog, frog, horse, ship and truck. It was divided to $5,000$ randomly selected images per class as training set and the rest, as testing samples. The second subset is called CIFAR-100 having 600 images in each of 100 classes. These classes also come in 20 super-classes of five class each. The SVHN dataset Netzer et al. (2011) was extracted from a large number of Google Street View images by automated algorithms and the Amazon Mechanical Turk (AMT) framework. It consists of over $600,000$ labeled characters in full numbers and MNIST-like cropped digits in $32 \times 32$. Three subsets are available containing $73,257$ digits for training, $26,032$ for testing and $531,131$ extra samples.

We consider two different scenarios to evaluate the performance of our distributed transfer learning algorithm. In the first experiment, we try to observe the performance of fine-tuning for pairs

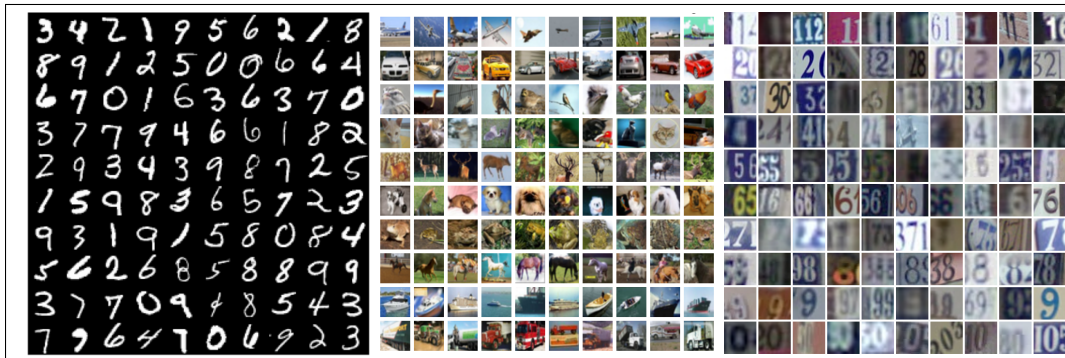

Figure 2: Examples of MNIST, CIFAR and SVHN Datasets

of datasets with close data distributions or number of classes. We select MNIST & SVHN and CIFAR-10 & CIFAR-100 as original-target domains and report the transfer learning results in form of train-test errors. In the second experiment, we apply transfer learning for pairs of datasets with far data/class setups which are MNIST & CIFAR-10 and SVHN & CIFAR-100. In this experiment, we arrange the datasets to examine the effect of dissimilar distributions rather than overfitting.

Before moving forward to discuss the experiments, we report the baseline train-test errors for the datasets in Table 1. These results are produced by the deep learning library provided by the Oxford Visual Geometry Group Vedaldi & Fulkerson (2008).

Table 1: Baseline Performances of Deep Learning

|  | Baseline | |
|---|---|---|
|  | **Train Error (%)** | **Test Error (%)** |
| **MNIST** | 0.04 | 0.55 |
| **SVHN** | 0.13 | 3.81 |
| **CIFAR-10** | 0.01 | 19.40 |
| **CIFAR-100** | 0.17 | 50.90 |

## 3.1 EXPERIMENT 1

Table 2 shows the performance of conventional and distributed transfer learnings for the first scenario. The first values before dash correspond to the training errors (left) and the second ones present the testing errors (right).

In this experiment, we target two pairs of datasets (original-target domains) which contain similar data and perform number/object recognition tasks. We report the results for both conventional and our distributed transfer learning methods. By conventional Bengio et al. (2012), we mean training the original dataset and fine-tuning of the target one. With distributed, we aim at training the original dataset but employing the basic probability assignment for the transfer learning.

It can be seen that the results for the conventional transfer learning follows our argument on size of network and number of model parameters Sermanet et al. (2013). Compared to Table 1, MNIST does a poor job on transferring of SVHN due to the overfitting of SVHN over MNIST network. In contrast, SVHN perform quite well on transferring MNIST.

Table 2: Performance of Conventional and Distributed Transfer Learning for Experiment 1

| | | Target | |
|---|---|---|---|
| | **Conventional** | **MNIST** | **SVHN** |
| **Original** | **MNIST** | - | 0.01 — 29.57 |
| | **SVHN** | 0.35 — 1.04 | - |

| | | Target | |
|---|---|---|---|
| | Distributed | **MNIST** | **SVHN** |
| **Original** | **MNIST** | - | 0.24 — 5.18 |
| | **SVHN** | **0.16 — 0.46** | - |

| | | Target | |
|---|---|---|---|
| | Conventional | **CIFAR-10** | **CIFAR-100** |
| **Original** | **CIFAR-10** | - | 0.53 — 68.44 |
| | **CIFAR-100** | 0.11 — 24.08 | - |

| | | Target | |
|---|---|---|---|
| | Distributed | **CIFAR-10** | **CIFAR-100** |
| **Original** | **CIFAR-10** | - | 0.29 — 54.32 |
| | **CIFAR-100** | **0.05 — 18.24** | - |

On the other hand, transferring of SVHN from MNIST does not overfit when our distributed transfer learning is employed. In both settings of original-target domains, our distributed strategy outperforms the conventional transfer learning approach.

The experiment on CIFAR pair exposes more interesting results due to the fact that both datasets have the same number of samples but completely different distributions among the classes. In practice, CIFAR-100 includes all the classes of CIFAR-10 but CIFAR-10 does not have any clue of the several classes of CIFAR-100. The conventional experiments show that CIFAR-10 transfers well on CIFAR-100 but it cannot perform transferring although the target network does not overfit.

All in all, the performance of our distributed transfer learning (bold values) is better than the conventional scheme and also, outperforms the baseline deep learning practices.

### 3.2 Experiment 2

In Table 3, we reports the results for both conventional and distributed transfer learnings on the second scenario. Here, we pair datasets such that the similarity of their data distributions and number of classes get minimized and they are originally trained for different tasks. It is obvious that our distributed transfer learning outperforms all the conventional results.

For the first setup, CIFAR-10 does a better transfer learning than MNSIT although the number of classes are the same. It seems that CIFAR-10 provides better generalization due to higher diversity among its classes. Here, our distributed algorithm performs better than the conventional process and,

Table 3: Performance of Conventional and Distributed Transfer Learning for Experiment 2

|  | | Target | |
| --- | --- | --- | --- |
| | Conventional | **MNIST** | **CIFAR-10** |
| **Original** | **MNIST** | - | 0.43 — 28.92 |
| | **CIFAR-10** | 0.44 — 2.37 | - |

|  | | Target | |
| --- | --- | --- | --- |
| | Distributed | **MNIST** | **CIFAR-10** |
| **Original** | **MNIST** | - | 0.25 — 20.85 |
| | **CIFAR-10** | 0.23 — 0.95 | - |

|  | | Target | |
| --- | --- | --- | --- |
| | Conventional | **SVHN** | **CIFAR-100** |
| **Original** | **SVHN** | - | 0.71 — 89.31 |
| | **CIFAR-100** | 0.01 — 12.18 | - |

|  | | Target | |
| --- | --- | --- | --- |
| | Distributed | **SVHN** | **CIFAR-100** |
| **Original** | **SVHN** | - | 0.46 — 61.10 |
| | **CIFAR-100** | 0.28 — 7.25 | - |

targeting of MNIST on CIFAR-10 network gives close performance to the deep learning outcomes. The second setup leads to the overfitting of SVHN over CIFAR-100 network due to huge number of samples. The other outcome is the poor performance of transferring CIFAR-100 over SVHN network as a result of huge conceptual gap between original-target domains.

Our observations show that fine-tuning on training set and calculating BPA on validation, result in better generalization of the transferred model on testing set. On the other hand, computing of BPA on training plus validation sets gives higher performance in case of hugely different number of classes in original-target datasets. Since we employ BPA to address the class-imbalance problem, we reckon that it better captures the distribution of data by adjoining both train/validation sets especially when we intend to transfer few classes of original dataset to the larger number of classes in the target.

## 4 CONCLUSION

We introduce a novel transfer learning for deep convolutional networks that tackles the optimization complexity of a highly non-convex objective by breaking it to several distributed fine-tuning operations. This also resolves the imbalance class coverage between original-target domains by using basic probability assignment across several week single-filter classifiers. By the above boosting, the overall performance shows considerable improvement over conventional transfer learning scheme. We conduct several experiments on publicly available datasets and report the performance as train-test errors. The results confirm the advantage of our distributed strategy for the transfer learning.

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
