# Peer review of "Distributed Transfer Learning for Deep Convolutional Neural Networks by Basic Probability Assignment"

_ICLR 2017 — rejected_

[Official Review · AnonReviewer2 · rating 4 · confidence 4 · 16 Dec 2016]
**No Title**

This paper proposed to use the BPA criterion for classifier ensembles.

My major concern with the paper is that it attempts to mix quite a few concepts together, and as a result, some of the simple notions becomes a bit hard to understand. For example:

(1) "Distributed" in this paper basically means classifier ensembles, and has nothing to do with the distributed training or distributed computation mechanism. Granted, one can train these individual classifiers in a distributed fashion but this is not the point of the paper.

(2) The paper uses "Transfer learning" in its narrow sense: it basically means fine-tuning the last layer of a pre-trained classifier.

Aside from the concept mixture of the paper, other comments I have about the paper are:

(1) I am not sure how BPA address class inbalance better than simple re-weighting. Essentially, the BPA criteria is putting equal weights on different classes, regardless of the number of training data points each class has. This is a very easy thing to address in conventional training: adding a class-specific weight term to each data point with the value being the inverse of the number of data points will do.

(2) Algorithm 2 is not presented correctly as it implies that test data is used during training, which is not correct: only training and validation dataset should be used. I find the paper's use of "train/validation" and "test" quite confusing: why "train/validation" is always presented together? How to properly distinguish between them?

(3) If I understand correctly, the paper is proposing to compute the BPA in a batch fashion, i.e. BPA can only be computed when running the model over the full train/validation dataset. This contradicts with the stochastic gradient descent that are usually used in deep net training - how does BPA deal with that? I believe that an experimental report on the computation cost and timing is missing.

In general, I find the paper not presented in its clearest form and a number of key definitions ambiguous.

[Official Review · AnonReviewer1 · rating 3 · confidence 3 · 17 Dec 2016 (modified: 23 Jan 2017)]

Update: I thank the author for his comments! At this point, the paper is still not suitable for publication, so I'm leaving the rating untouched.

This paper proposes a transfer learning method addressing optimization complexity and class imbalance.

My main concerns are the following:

1. The paper is quite hard to read due to typos, unusual phrasing and loose use of terminology like “distributed”, “transfer learning” (meaning “fine-tuning”), “softmax” (meaning “fully-connected”), “deep learning” (meaning “base neural network”),  etc. I’m still not sure I got all the details of the actual algorithm right.

2. The captions to the figures and tables are not very informative – one has to jump back and forth through the paper to understand what the numbers/images mean.

3. From what I understand, the authors use “conventional transfer learning” to refer to fine-tuning of the fully-connected layers only (I’m judging by Figure 1). In this case, it’s essential to compare the proposed method with regimes when some of the convolutional layers are also updated. This comparison is not present in the paper.

Comments on the pre-review questions:

1. Question 1: If the paper only considers the case |C|==|L|, it would be better to reduce the notation clutter.

2. Question 2: It is still not clear what the authors mean by distributed transfer learning. Figure 1 is supposed to highlight the difference from the conventional approach (fine-tuning of the fully-connected layers; by the way, I don’t think, Softmax is a conventional term for fully-connected layers). From the diagram, it follows that the base CNN has the same number of convolutional filters at every layer and, in order to obtain a distributed ensemble, we need to connect (for some reason) filters with the same indices. This does not make a lot of sense to me but I’m probably misinterpreting the figure. Could the authors revise the diagram to make it clearer?

Overall, I think the paper needs significant refinement in order improve the clarity of presentation and thus cannot be accepted as it is now.

[Official Review · AnonReviewer3 · rating 3 · confidence 4 · 22 Dec 2016]

This work proposes to use basic probability assignment to improve deep transfer learning. A particular re-weighting scheme inspired by Dempster-Shaffer and exploiting the confusion matrix of the source task is introduced. The authors also suggest learning the convolutional filters separately to break non-convexity. 

The main problem with this paper is the writing. There are many typos, and the presentation is not clear. For example, the way the training set for weak classifiers are constructed remains unclear to me despite the author's previous answer. I do not buy the explanation about the use of both training and validation sets to compute BPA. Also, I am not convinced non-convexity is a problem here and the author does not provide an ablation study to validate the necessity of separately learning the filters. One last question is CIFAR has three channels and MNIST only one: How it this handled when pairing the datasets in the second set of experiments?

Overall, I believe the proposed idea of reweighing is interesting, but the work can be globally improved/clarified. 

I suggest a reject.

[Final Decision · Program Chairs · 06 Feb 2017]
**ICLR committee final decision**

All three reviewers appeared to have substantial difficulties understanding the proposed approach due to unclear presentation. This makes it hard for the reviewers to evaluate the originality and potential merits of the proposed approach, and to assess the quality of the empirical evaluation. I encourage the authors to improve the presentation of the study.